# Differences in Body Composition among Patientsafter Hemorrhagic and Ischemic Stroke

**DOI:** 10.3390/ijerph17114170

**Published:** 2020-06-11

**Authors:** Jacek Wilczyński, Marta Mierzwa-Molenda, Natalia Habik-Tatarowska

**Affiliations:** Laboratory of Posturology, Collegium Medicum, Jan Kochanowski University in Kielce, 25-516 Kielce, Poland; martamierzwa@wp.pl (M.M.-M.); habiknatalia@gmail.com (N.H.-T.)

**Keywords:** body composition, hemorrhagic stroke, ischemic stroke

## Abstract

The aim of the study was to assess differences in the body composition of patients after hemorrhagic and ischemic stroke. There were 74 male participants in the study, of which 13 (18%) experienced hemorrhagic stroke, while 61 (82%) were after ischemic stroke. Significantly (*p* < 0.05) higher values of body composition variables were noted for ischemic compared to hemorrhagic strokes, and concerned: body mass (BM) (kg), basal metabolic rate (BMR) (kJ), fat-free mass (FFM) (kg), total body water (TBW) (kg), muscle mass (MM) (kg), visceral fat level (VFL), bone mass (BoM) (kg), extracellular water(ECW) (kg),intracellular water (ICW) (kg), trunk fat-free mass (TFFM) (kg) and trunk muscle mass (TMM) (kg)in the paretic upper limb; FFM (kg) and MM (kg) in the non-paretic upper limb; FFM (kg) and MM (kg) in the paretic lower limbas well as FFM (kg) and MM (kg) in the non-paretic lower limb without paresis. Only for the variables fat mass (FM) (kg), body mass index (BMI), metabolic age (MA), trunk fat mass (TFM) (kg), and FM (kg) in the paretic upper limb and FM (kg) in the non-paretic upper limb were there no significant differences. Significant differences in body composition of patients after hemorrhagic and ischemic stroke have been demonstrated. Individuals after ischemic stroke had significantly worse body composition. Incorrect body composition is a significant risk factor, especially of ischemic stroke.

## 1. Introduction

According to the 2016 report by the World Stroke Organization (WSO), 13.7 million people worldwide are diagnosed with stroke every year. Among this group, over five million die and five million remain disabled [1,2]. Approximately 60,000 new cases of stroke are registered in Poland annually. Incidence rates in our country are 177/100,000 for men and 125/100,000 for women and are worse than in Western Europe, Japan, or the United States [3]. The mortality rate in Poland is 106/100,000 for men and 79/100,000 for women, which is one of the highest in Europe [4,5].

Until 2013, stroke was defined by the World Health Organization as rapidly developing clinical signs of focal (or global) brain function disorder lasting over 24 h or leading to death, for no apparent cause other than that of vascular origin [6]. In 2013, Sacco et al. from the American Heart Association and American Stroke Association updated this definition because of the progress that has been made in both the possibilities of research and the treatment itself. Currently, stroke is defined as a neurological deficit associated with acute focal damage to the central nervous system caused by vascular disorders including cerebral ischemia, intrace rebral hemorrhage and subarachnoid hemorrhage. The new definition makes it possible to recognize stroke even when focal neurological symptoms persist for a shorter period of time (e.g., in a patient whose deficits resolved under the influence of thrombolytic therapy or in the event of unambiguous confirmation of an ischemic focus using neuroimaging examination [7,8,9,10]).

Stroke, just after coronary heart disease, is the second leading cause of death and the first cause of disability in the world. Most often, it concerns patients above the age of 65. A disturbing, steady increase in incidence is observed in people over 40 who, in addition to acquired disability as a result of stroke, are excluded from professional activity [11,12,13,14,15].

Accordingly, the World Health Organization and World Stroke Organization recommend expanding efforts to raise awareness of population risk factors for stroke as well as implementing effective prevention strategies at global, regional, and national levels. Created by the European Parliament in 2004, the Stroke Alliance for Europe recommends primary and secondary prevention, improvement of health care, rehabilitation, and assessment of quality of life [16,17]. According to these recommendations, secondary prevention should consist of detecting risk factors and combating them through lifestyle changes and appropriate pharmacological management [18]. Modifiable risk factors for stroke are hypertension, diabetes, ischemic heart disease, atrial fibrillation, dyslipidemia, stimulant intake, metabolic and body composition disorders as well as obesity [19]. These indicators can be improved by intensifying care in acute stroke as well as by actively decreasing risk factors.

Abnormal body composition, high systolic blood pressure, BMI, cholesterol, and diabetes represent 70% of stroke risk factors [20]. Behavioral factors such as smoking, poor diet, and low physical activity or environmental aspects (i.e., air pollution) also play an important role in increasing the risk of stroke [21].

Evaluation of body composition is an important factor in determining health. It plays a significant role, especially when analyzing the content of body fat in cases of eating or hormonal disorders. Knowledge of ontogenetic variability in body composition contributes to a better understanding of physiological and biochemical processes occurring in the body. In body composition, we may distinguish lean body mass (LBM), also called lean or adipose tissue. It comprises 70–85% of total body mass, in which connective tissue and bones cover 15%. In contrast, adipose tissue covers from 10 to 20% of an adult’s body mass. Excess body fat, caused by metabolic disorders, a sedentary lifestyle, and excessive supply of food products in relation to energy needs leads to obesity. This, in turn, carries a risk of stroke, heart and circulatory system diseases, diabetes, and/or cancer. Body composition disorders leading to obesity are a risk factor for the development of hypertension and diabetes mellitus and their complications play an important indirect role in stroke epidemiology. Moreover, this condition is associated with the activity of strong cytokines, affecting the activity of the sympathetic nervous system, the renin–angiotensin axis, endothelial function, and microcirculation [22,23,24]. In the authors’ research, it was hypothesized that people after ischemic stroke had significantly worse body composition compared to those following hemorrhagic stroke and are therefore more likely to experience stroke. The aim of the study was to assess the differences in the body composition of patients after hemorrhagic and ischemic stroke.

## 2. Materials and Methods

The study comprised 74 males, 13 (18%) of them after hemorrhagic stroke and 61 (82%) following ischemic stroke. The most numerous group included those pharmacologically treated, among which there were 62 (83%) patients. Thrombolytic treatment with IV tPA and mechanical throm bectomy was received by 16% of patients. There were 42 (57%) men with left- and 32 (43%) with right-sided paresis. The research was conducted in the years 2018–2019 at the Neurological Rehabilitation Ward of the Provincial Integrated Hospital in Kielce (Poland). Written consent to participate in the study was obtained from each subject. Each participant of the study was informed about its purpose and the possibility of withdrawing from participation at any stage. The examined subjects were diagnosed by doctors: a neurosurgeon, radiologist, and neurologist with reference to stroke. The research was approved by the Bioethics Committee at the Faculty of Medicine and Health Sciences of Jan Kochanowski University in Kielce (No. 33/2016).

The subjects were qualified for examination by a neurologist who evaluated the patient’s clinical condition and their ability to assume a standing position. Inclusion criteria were logical contact with the patient and the ability to maintain standing position for at least 120s. The study took place an average of 10 days after the onset of the stroke.

Body composition was measured using bioelectrical impedance analysis (BIA) with the Tanita MC-780 multi-frequency segment body composition analyzer. The non-invasive method of diagnostic examination allowed for the analysis of body composition using the electrical resistance of the body’s tissues, so-called impedance. The Tanita MC-780 segmented body composition analyzer implements innovative multi frequency technology (i.e., currents with variable frequencies: 5, 50, 250 and 500 kHz). This prohibited assessment of tissue resistance and conductivity. The flow of alternating currents was possible due to different fluid contents in the tissues. Body composition analysis of individual segments (upper and lower limbs, trunk) was shown by taking into account the body side (right and left) and the distinction between tissue reactance and resistance (muscle, fat, visceral tissue). The patient was examined in a standing position, the feet (located on the base of the analyzer) were in contact with the built-in electrodes. Patient data (age, gender, body height) were entered by the investigator. During the first stage, body mass was determined; in the following stage (patient’s hands, on the handles with built-in electrodes), impedance was measured. Full segmented analysis was carried out in 30 s. Values of segmental measurements for the lower and upper limbs as well as the trunk are indicated in specific SI units: (kg), (kJ), (%), (°), (Ω), and (kg/m²). The body analyzer has a certificate approving its use in medical fields. Within the overall analysis, the following body composition parameters were examined: body mass (BM) (kg), basal metabolic rate (BMR) (kJ), fat percentage (FP) (%), fat mass (FM) (kg), fat-free mass (FFM) (kg), total body water (TBW) (kg), muscle mass (MM) (kg), impedance (IMP) (Ω), body mass index (BMI) (kg/m²), visceral fat level (VFL), bone mass (BoM) (kg), extracellular water (ECW) (kg), intracellular water (ICW) (kg), metabolic age (MA) (years) and phase angle (PA) (°). For the trunk and each paretic and non-paretic limb, the following were analyzed: FP (%), FM (kg), FFM (kg) and muscle mass (MM) (kg). BM (kg) means the total weight of the body in kilograms. BMR (kJ) determines the minimal level of energy that the body needs to be able to function properly in resting state. FP (%) is the percentage of body fat in total body mass. FM (kg) is the mass of fat tissue in kilograms. FFM (kg) is the most important metabolic component of the body. It consists of muscle, internal organ, and bone mass. TBW (kg) is the total fluid content of the body. MM (kg) includes skeletal and smooth muscle mass. IMP (Ω) is a generalization of electrical resistance. The electrical signal easily flows through tissues containing large amounts of water (e.g., muscle tissue). Passing through adipose tissue, it encounters greater resistance because it contains small amounts of fluid. This resistance is called impedance. Impedance readings are then entered into medically established mathematical formulas to calculate body composition. BMI (kg/m²) is the ratio of body mass to body height, being a standard indicator.VFL means the visceral fat located deep in the abdominal area. BM (kg) is the mass of bone minerals in the body. ECW (kg) denotes its content in the body. ICW (kg) is its content in the body. MA is the metabolic rate. Genes, diet, physical activity as well as hormonal balance affect metabolic age. In addition, this index also depends on basal metabolic rate, body water content, body fat mass, and muscle mass. PA (°) determines overall health and functional quality of the tissues and cells. It is also an indicator of the body’s nutritional level. It is calculated from the resistance and reactance coefficient. Low phase angle may indicate physical exhaustion. It is also characteristic for malnourished and chronically ill individuals. The correct phase angle for women is > 5°, while for men, it should be > 6°. We compared the body composition of people after hemorrhagic and ischemic stroke.

## 3. Applied Statistical Methods

Statistical analysis was performed using the following tools: Visual Studio Code, MS Excel, and Spyder. Statistical libraries in Python-language were used for the calculations. The data were verified for normality of distribution and homogeneity using the Shapiro-Wilk normality test and Levene’s test of homogeneity of variance. First, the mean, standard deviation, minimum, and maximum ranges for body composition in the group of hemorrhagic and ischemic strokes were calculated. Next, it was checked whether the above-mentioned groups significantly differed in terms of body composition using ANOVA analysis of variance.

## 4. Results

The values of location and dispersion measures for the discussed variable body composition had varied distribution of variables both in men with hemorrhagic and ischemic stroke.

Detailed differences in body composition between hemorrhagic and ischemic stroke were as follows: BM (kg), BMR (kJ), FFM (kg), TBW (kg), ECW and ICW (kg), MM (kg),VFL and BoM were all significantly higher in those after ischemic stroke (*p* < 0.001); while PA (°), FP (%), and) IMP (Ω) were significantly higher in those after hemorrhagic stroke (*p* < 0.001). In the case of FM (kg) (*p* = 0.099), MA (*p* = 0.566) and BMI (*p* = 0.309), no significant differences were noted.

Within the trunk, trunk fat percentage TFP (%) was significantly higher in patients after hemorrhagic stroke (*p* = 0.034), while trunk fat-free mass (TFFM) (kg) (*p* < 0.0001) and trunk muscle mass (TMM) (kg) (*p* < 0.001) were significantly higher in those after ischemic stroke. For TFM (kg), no significant differences were found (*p* = 0.618).

Within the paretic upper limb, fat percentage (%) was significantly higher in patients following hemorrhagic stroke (*p* < 0.0001) while for FFM (kg) and MM (kg), their values were noted to be significantly higher in the case of ischemic stroke (*p* < 0.001). FM (kg) did not demonstrate any significant differences (*p* = 0.083).

Within the non-paretic upper limb, FP (%) was significantly higher in patients after hemorrhagic stroke (*p* < 0.001) while for FFM (kg) and MM (kg), the obtained values were significantly higher in individuals following ischemic stroke (*p* < 0.001).Once again, fat mass (kg) did not indicate any significant differences (*p* = 0.091).

In the lower paretic limb, FP (%) and FM (kg) were noted as significantly higher in patients after hemorrhagic stroke (*p* < 0.001), while FFM (kg) as well as MM (kg) had a significantly higher level in individuals following ischemic stroke (*p* < 0.001).

Within the non-paretic lower limb, FP (%) and FM (kg) were significantly higher in patients after hemorrhagic stroke (*p* < 0.001) while values for FFM (kg) and MM (kg) were significantly higher in patients following ischemic stroke (*p* < 0.001). These results are presented in Table 1.

## 5. Discussion

In our research, significant differences in body composition between hemorrhagic and ischemic stroke were demonstrated. Individuals with ischemic stroke had significantly worse body composition. Incorrect body composition is an important risk factor of stroke, especially the ischemic-type. In stroke prevention, much depends on the patients themselves. It is important to be aware that there are certain factors affecting the onset of stroke.

Obesity, high VFL, FP (%), FM (kg) and low PA occurred in both groups. Low phase angle means the inability of cells to store energy and the incorrect permeability of cell membranes. It may also indicate physical exhaustion. Phase angle measured in a non-invasive manner using bioelectrical impedance analysis is a potentially new, objective, and useful indicator of proper nutritional status that may be used in clinical practice.

In the published research, it is stated that obesity is strongly and positively correlated with ischemic stroke, mainly through its effect on blood pressure. However, no relationship was found between obesity and hemorrhagic stroke within the normal BMI range (< 25 kg/m^2^), nonetheless, a correlation between hemorrhagic stroke and below-normal BMI was indicated. As ischemic stroke concerns the majority of stroke cases, findings of intrace rebral hemorrhage should not detract from the fundamental importance of high obesity as the main modifiable determinant of all strokes [25].

In turn, in a different study, body mass index was a significant risk factor for total and ischemic stroke cases in both men and women. However, abdominal obesity was a risk factor for hemorrhagic and ischemic stroke only in men [26]. In other research, it has been shown that women have a higher BMI associated with an increased risk of ischemic stroke while, at the same time, being at a reduced risk of hemorrhagic stroke. All available published evidence indicates that the risk associated with high BMI is greater in ischemic stroke [27].

The purpose of the next study was to assess the relationship between obesity through fat imaging and stroke outcomes in patients with acute ischemic stroke subjected to intravenous throm bolitic therapy. The results of this study indicate that low visceral abdominal fat proportion is associated with favorable and excellent outcomes in acute ischemic stroke patients treated with intravenous thrombolysis. Better clinical outcomes in obese patients were also associated with a lower proportion of VAT [28].

It is still unclear whether high absolute FM increases the risk of stroke independently. Other authors have examined the correlation between FM and silent brain infarction/white matter change (SI/WMC) using computed tomography. A target population that had never been diagnosed with stroke or dementia was randomly selected. FM was measured by bioelectrical impedance analysis (BIA). The subjects were divided into three groups according to the FM (gender-specific tertiles (GTx). The findings suggest that high FM may be an independent risk factor for ischemic stroke among adults free from stroke and dementia, especially in women [29], which was confirmed in the authors’ study.

In the published research, the importance of BMI as a risk factor for both ischemic and hemorrhagic stroke is highlighted [30]. However, VFL is a stronger predictor of stroke susceptibility than BMI. In the research conducted by the authors, this was also confirmed. Nevertheless, in yet a different study, it was shown that in men, BMI was the only indicator significantly associated with stroke, while in women, the same regarded waist-to-hip ratio [31]. The results of another study [32], similar to the case of the authors’, emphasized a significant correlation between stroke and abdominal fat.

Additionally, clinical situations involving the excess or deficiency of water can have extremely serious consequences for the functioning of the body and subsequent stroke-related disorders. In the following cited work, a group of patients were examined with the use of the BIA apparatus three times: on the first, seventh, and tenth day of hospital stay. This was done in order to determine the level of body hydration. In a comparative analysis of data, it was shown that on the first day of hospitalization, all the measurements of electrical bio-impedance parameters in the patients were significantly different from those obtained in the control group. In the case of TBW, ECW and ICW, the patients’ scores were significantly higher than those of the control group. Only with respect to the over-hydration index did the patients achieve a significantly lower score than the control group. Assessment of hydration status in patients with diagnosed stroke indicated slight dehydration in relation to the control group, but falling within the scope of normo volemia, according to bio-impedance measurement standards [33].

The association between obesity and stroke remains controversial, with earlier studies suggesting that differences might stem from heterogeneous stroke subtype compositions. BMI is a risk factor for both ischemic and hemorrhagic stroke, but shows different relationships with each. When the total burden of stroke is considered, there is an urgent need to find better ways of reducing the trend toward growing obesity.

Secondary prevention of strokes should be based on the detection of risk factors and their eradication by lifestyle changes as well as appropriate pharmacological and physiotherapy treatment. Striving to eliminate risk factors associated with body composition can positively affect the incidence of stroke and its course. Lifestyle modification in the form of regular physical exertion, a proper diet with the introduction of eating habits limiting the consumption of animal fats and salt, care for adequate body mass and composition as well as smoking cessation have great significant impact on the prevention of stroke. However, it should be borne in mind that there are non-modifiable factors for the occurrence of stroke (e.g., genetic conditions or age). There are few studies comparing the body composition of patients following hemorrhagic and ischemic stroke using the modern method of bio-impedance. The strength of the authors’ study was the BMR, VFL, and PA assessment as well as division into holistic, trunk-, and limb-related analysis. A weaker point of the research was limiting the study only to a group of men. Nonetheless, the results of this study provide great substantive value.

## 6. Conclusions

Significant differences in the body composition of patients after hemorrhagic and ischemic stroke have been demonstrated. Individuals after ischemic stroke had significantly worse body composition and significantly higher values of body composition variables compared to hemorrhagic stroke patients concerning: BM, BMR (kJ), FFM (kg), TBW (kg), MM (kg), VFL, BoM (kg), ECW (kg), ICW (kg), TFFM (kg), TMM (kg), FFM (kg) and MM (kg) in the upper limb with paresis; FFM (kg) and MM (kg) in the upper non-paretic limb; FFM (kg) and MM (kg) in the lower limb with paresis; and FFM (kg) and MM (kg) in the lower limb without paresis. Incorrect body composition is a significant risk factor, especially of ischemic stroke.

## Figures and Tables

**Table 1 ijerph-17-04170-t001:** Differences in body composition between hemorrhagic and ischemic stroke.

Body Composition	Hemorrhagic Stroke		Ischemic Stroke		ANOVA F	*p*
	X	SD	X	SD		
Age	59.5	9.0	61.7	10.4	0.50	0.483
Body height (cm)	169.8	8.4	172.2	6.5	38.95	0.001
Body mass (BM) (kg)	81.4	16.4	82.9	15.4	11.91	0.001
Basal metabolic rate (BMR) (kJ)	7319.9	1066.4	7579.4	1074.3	59.42	0.001
Fat percentage (FP) (%)	25.1	7.4	23.8	7.2	30.76	0.001
Fat mass (FM) (kg)	21.4	10.1	20.6	9.1	2.78	0.099
Fat-free mass (FFM) (kg)	60.0	8.1	62.4	8.1	83.76	0.001
Total body water (TBW) (kg)	42.3	5.9	43.8	6.4	67.52	0.001
Muscle mass (MM) (kg)	57.0	7.7	59.3	7.7	83.55	0.001
Impedance (IMP) (Ω)	542.3	73.5	534.1	82.8	29.06	0.001
Body mass index (BMI) (kg/m²)	28.3	5.9	28.0	5.0	1.05	0.309
Visceral fat level (VFL)	12.2	4.4	12.9	4.1	21.35	0.001
Bone mass (BoM) (kg)	3.0	0.4	3.1	0.4	87.40	0.001
Extracellular water (ECW) (kg)	18.4	2.4	18.9	2.2	55.06	0.001
Intracellular water (ICW) (kg)	24.0	3.7	24.9	4.4	67.34	0.001
Metabolic age (MA)	56.1	10.2	56.1	11.7	0.33	0.566
Phase angle (PA) (°)	6.2	0.6	6.1	1.4	8.68	0.004
Trunk fat percentage (TFP) (%)	24.6	7.8	24.1	8.2	91.37	0.034
Trunk fat mass (TFM) (kg)	11.8	5.3	11.9	5.5	23.43	0.618
Trunk fat-free mass (TFFM) (kg)	34.1	3.7	35.3	3.9	58.07	0.001
Trunk predicted muscle mass (TPMM) (kg)	32.4	3.6	33.6	3.8	68.72	0.001
Fat percentage (FP) (%) in paretic upper limb	23.2	9.2	21.1	6.2	86.45	0.001
Fat mass (FM) (kg) in paretic upper limb	1.2	0.8	1.1	0.6	22.50	0.083
Fat-free mass (FFM) (kg) in paretic upper limb	3.2	0.7	3.7	0.7	71.72	0.001
Muscle mass (MM) (kg) in paretic upper limb	3.1	0.4	3.5	0.5	54.14	0.001
Fat percentage (FP) (%) in non-paretic upper limb	23.4	9.3	21.2	6.8	37.12	0.001
Fat mass (FM) (kg) in non-paretic upper limb	1.3	0.9	1.2	0.6	3.08	0.091
Fat-free mass (FFM) (kg) in non-paretic upper limb	3.3	0.6	3.8	0.8	75.34	0.001
Muscle mass (MM) (kg) in non-paretic upper limb	3.3	0.7	3.5	0.7	68.72	0.001
Fat percentage (FP) (%) in paretic lower limb	25.2	9.4	23.7	7.8	32.55	0.001
Fat mass (FM) (kg) in paretic lower limb	3.6	2.2	3.1	1.7	2.92	0.001
Fat-free mass (FFM) (kg) in paretic lower limb	9.4	1.8	10.1	1.8	52.03	0.001
Muscle mass (MM) (kg) in paretic lower limb	9.1	1.7	9.6	1.6	54.14	0.001
Fat percentage (FP) (%) in non-paretic lower limb	25.6	9.3	24.3	7.8	4.62	0.001
Fat mass (FM) (kg) in non-paretic lower limb	3.7	2.2	3.2	1.4	0.25	0.001
Fat-free mass (FFM) (kg) in non-paretic lower limb	9.6	1.8	9.9	1.6	89.17	0.001
Muscle mass (MM) (kg) in non-paretic lower limb	9.0	1.7	9.4	1.5	88.85	0.001

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
