# Peer review of "Differences in Body Composition among Patientsafter Hemorrhagic and Ischemic Stroke"

_ijerph, 2020, doi:10.3390/ijerph17114170_

Round 1

Reviewer 1 Report

I was honored to review the manuscript entitled "Differences in body composition among subjects after hemorrhagic and ischemic stroke" submitted to International Journal of Environmental Research and Public Health.

.

Taking into account the multiple studies ongoing in this field this type of study is needed.  I have only few small remarks that authors should adress properly.

I recommend to accept the manuscript after minor revision.

There are only some points to correct:

 - please provide clearly the aim of this study

 - please provide the list of abbreviations.

 - please provide the number of Ethical Approval

 - In my opinion discussion is too short, so I suggest that Introduction and Discussion section needs improvement- please cite doi: 10.3967/bes2016.095 ; 10.20452/pamw.4482 ; 10.1039/c9fo01006h

 - In discussion please provide “study strong points” and “study limitation” section.

I recommend to accept the manuscript after minor revision.

Author Response

Answers to the comments of the First Reviewer

Thank you very much for your insightful comments and recommendations. We are very grateful for the time you have devoted to the review. We have implemented the required changes in the text. Below, please find a point-by-point response to the comments. Once more, thank you for the insight which, we are sure, has significantly improved the substantive value of our manuscript.

  1. I was honored to review the manuscript entitled "Differences in body composition among subjects after hemorrhagic and ischemic stroke" submitted to International Journal of Environmental Research and Public Health. Taking into account the multiple studies ongoing in this field this type of study is needed.  I have only few small remarks that authors should adress properly. I recommend to accept the manuscript after minor revision.
  2. There are only some points to correct:
  • please provide clearly the aim of this study

Response: The aim of the study has been clearly stated: The aim of the study was to assess the differences in body composition among individuals after hemorrhagic and ischemic stroke.

  • please provide the list of abbreviations.

Response: A list of the implemented abbreviations has been provided and all of them have been explained in the text of the article.

  • please provide the number of Ethical Approval

Response: The number of the Ethical Committee’s approval has been provided: The research was approved by the Bioethics Committee at the Faculty of Medicine and Health Sciences of Jan Kochanowski University in Kielce (No. 33/2016).

  1. In my opinion discussion is too short, so I suggest that Introduction and Discussion section needs improvement- please cite doi: 10.3967/bes2016.095 ; 10.20452/pamw.4482 ; 10.1039/c9fo01006h

Response: The ‘Discussion’ section has been corrected and elaborated. In the ‘Introduction’, the following additional works have been cited:

  • SzuliÅ„ska M, Skrypnik D, Ratajczak M, Karolkiewicz J, Madry E, Musialik K, Walkowiak J, Jakubowski H, BogdaÅ„ski P. Effects of Endurance and Endurance-strength Exercise on Renal Function in Abdominally Obese Women with Renal Hyperfiltration: A Prospective Randomized Trial. Biomed Environ Sci. 2016; 29 (10): 706-712. doi: 10.3967/bes2016.095.
  • Tamowicz B, Mikstacki A, Urbanek T, Zawilska K. Mechanical methods of venous thromboembolism prevention: from guidelines to clinical practice. Pol Arch Intern Med. 2019; 31; 129(5): 335-341. doi: 20452/pamw.4482.
  • Skrypnik K, BogdaÅ„ski P, Sobieska M, Suliburska J. The effect of multistrain probiotic supplementation in two doses on iron metabolism in obese postmenopausal women: a randomized trial. Food Funct. 2019; 1; 10 (8): 5228-5238. doi: 1039/c9fo01006h.
  1. In discussion please provide “study strong points” and “study limitation” section.

Response:  In the ‘Discussion’ section, the strong points of the study have been given well as its limitations.

There are only a few studies comparing body composition of patients following hemorrhagic and ischemic stroke using the modern method of bioimpedance. The strength of our study is BMR, VFL and PA assessment as well as division into holistic, trunk and limb analysis. A weaker part of our research is limiting the study only to a group of men.                

  1. I recommend to accept the manuscript after minor revision.

Once more, we are exceptionally grateful for your in-depth review of our article. Your insight and comments will definitely allow for an increase in the substantive value of the manuscript. We hope that our detailed responses and the extensive changes to the text are sufficient for the publication of our text in your renowned journal. Thank you for your devoted time and effort.

Yours sincerely,

Assoc. Prof. UJK Jacek Wilczyński, Ph.D.

Reviewer 2 Report

The authors compared the body composition between subjects after hemorrhagic and ischemic stroke. However, there are several major flaws, made the ariticle inappropriate to publish.

1.The authors should read more scientific papers to follow the style of scentific writing and presentation.

2. Many statistical results, with shocking P values, can not be replicated.

Author Response

  • Answers to the comments of the Second Reviewer

Thank you very much for your insightful comments and recommendations. We are very grateful for the time you have devoted to the review. We have implemented the required changes in the text. Below, please find a point-by-point response to the comments. Once more, thank you for the insight which, we are sure, has significantly improved the substantive value of our manuscript.

  1. The authors compared the body composition between subjects after hemorrhagic and ischemic stroke. However, there are several major flaws, made the ariticle inappropriate to publish.
  2. The authors should read more scientific papers to follow the style of scentific writing and presentation.
  3. Many statistical results, with shocking P values, can not be replicated.

Response:

  • The article has been corrected. We have improved the style and presentation of our research results. In we ran more items for References.
  • The aim of the study has been clearly stated: The aim of the study was to assess the differences in body composition among individuals after hemorrhagic and ischemic stroke.
  • A list of the implemented abbreviations has been provided and all of them have been explained in the text of the article.
  • The number of the Ethical Committee’s approval has been provided: The research was approved by the Bioethics Committee at the Faculty of Medicine and Health Sciences of Jan Kochanowski University in Kielce (No. 33/2016).
  • The ‘Discussion’ section has been corrected and elaborated.
  • In the ‘Discussion’ section, the strong points of the study have been given well as its limitations.
  • The ‘Abstract’ section has been edited so that it is properly designed, containing the required conclusions.
  • In the ‘Introduction’ section, it has been explained as to why we have compared body composition among individuals with hemorrhagic and ischemic stroke.
  • We have explained the term "after stroke". Qualification for examination was performed by a neurologist. He evaluated the patient's clinical condition and ability to assume a standing position. The inclusion criteria for the study were logical contact with the patient and the ability to maintain a standing position for at least 120 seconds. On average, the test usually took place 10 days after the stroke.
  • It has been explained why body composition is of significance for risk of stroke.
  • This sentenced was corrected as follows: Eleven (13%) patients were treated via thrombolysis, while only one (4%) were treated using mechanical trombectomy.
  • The inclusion criteria were logical contact with the patient and the ability of each patient to maintain a standing position for at least 120 seconds. The study usually took place an average of 10 days following stroke onset. Patients were qualified for examination by a neurologist. He evaluated the patient's clinical condition and his ability to assume a standing position.
  • Tables 1 and 2 have been deleted. In the remaining table, SD for each IS and ICH has been added.
  • The title of the table has been changed to: Table 1. Differences in body composition between hemorrhagic and ischemic stroke.
  • In the ‘Methods’ section, we have explained the significance of all the examined variables of body composition. Body mass (kg) (BM) means the total weight of the body in kilograms. Basal Metabolic Rate (kJ) (BMR) determines the minimal level of energy that the body needs to be able to function properly in resting state. Fat Percentage (%) (FP) is the percentage of body fat in total body mass. Fat Mass (kg) (FM) is the mass of fat tissue in kilograms. Fat-free mass (FFM) (kg) is the most important metabolic component of the body. It consists of muscle, internal organ and bone mass. Total Body Water (kg) (TBW) is the total fluid content of the body. Muscle Mass (kg) (MM) includes skeletal and smooth muscle mass. Impedance (Ω) (IMP) is a generalisation of electrical resistance. The electrical signal easily flows through tissues containing large amounts of water, e.g. muscle tissue. Passing through adipose tissue it encounters greater resistance because it contains small amounts of fluid. This resistance is called impedance.Impedance readings are then entered into medically established mathematical formulas to calculate body composition. Body Mass Index (BMI) (kg/m²) is the ratio of body mass to body height. This is the standard body mass index. Visceral Fat Level (VFL) means visceral fat located deep in the abdominal area. Bone mass (BM) (kg) is the mass of bone minerals in the body. Extracellular Water (kg) (ECW) denotes its content in the body. Intracellular Water (ICW) (kg) is its content in the body. Metabolic Age (MA) means metabolic rate. Genes, diet, physical activity, as well as hormonal balance affect metabolic age. In addition, this index also depends on BMR, body water content, body fat mass, muscle mass. Phase angle (PA) (°) determines overall health and functional the quality of tissues and cells. It is also an indicator of the body's nutrition. It is calculated from the resistance and reactance coefficient. Low phase angle may indicate physical exhaustion. It is also characteristic for malnourished and chronically ill individuals. The correct phase angle for women is >5°, while for men, it should be >6°.
  • We have introduced the following terms: paretic limb and non-paretic limb. Once more, we have calculated the means and standard deviation for these variables as well as the Anova – F.
  • This has been corrected, and in the ‘Discussion’ section, we have applied the same abbreviations.
  • The ‘Discussion’ section has been extensively edited and elaborated. We have compared the results of our research with those obtained in other studies.
  • We have corrected the ‘Conclusions’ section and have applied the abbreviations.
  • Significant differences in the body composition among patients after hemorrhagic and ischemic stroke have been indicated in this study. Individuals who had experienced ischemic stroke demonstrated significantly worse body composition parameters. Improper body composition is an important stroke risk factor – especially with regard to ischemic incidences.

Once more, we are exceptionally grateful for your in-depth review of our article. Your insight and comments will definitely allow for an increase in the substantive value of the manuscript. We hope that our detailed responses and the extensive changes to the text are sufficient for the publication of our text in your renowned journal. Thank you for your devoted time and effort.

Yours sincerely,

Assoc. Prof. UJK Jacek Wilczyński, Ph.D.

Reviewer 3 Report

  1. Abstract has to be structured accordingly, because there is a lack of conclusions in it.
  2. In introduction the fact why are You targeting the body composition has to be accented.
  3. Also, You talk of risk factors for stroke, not the factors associated with poorer or better stroke outcome- as You state in the title: "AFTER hemorrhagic and ischemic stroke" so term after has to be explained- when? 10d-? 30d-? 180d-?
  4. You need to explain why the body composition is important for the stroke outcome more clearly.
  5. Line 65 -Thrombolysis was diagnosed in 11 patients (13%), while only one (4%) male was treated by mechanical thrombectomy.
  6. In methods- as this study is done in rehabilitation center, then You should point out, what time after stroke or ICH was this performed. 
  7. In results the two separate tables are not necessary- just us the 3rd table, but ad the SD for each IS and ICH.
  8. The title for 3rd table is incorrect. 
  9. In methods as You are measuring certain things (all that are in your results)- You should explain what they are and what they mean, and why You consider them important. 
  10. Also the results will differ if You will not use the left or right arm/leg instead i would suggest to use- paretic arms/legs nonparetic arms/legs, then You need to recalculate.
  11. As You describe the results You should use the terms and abbreviations, and then just the abbreviations, no need to explain the kg etc. in discussion.
  12. In discussion You should compare the results of Your study to other studies, not just tell what they have found, so You are not really discussing. 
  13. I don’t see an actual conclusion. Also here just use the abbreviations not the full text. 

Author Response

  • Answers to third reviewer's comments
  1. Abstract has to be structured accordingly, because there is a lack of conclusions in it.

Response: The ‘Abstract’ section has been edited so that it is properly designed, containing the required conclusions.

  1. In introduction the fact why are You targeting the body composition has to be accented.

Response: In the ‘Introduction’ section, it has been explained as to why we have compared body composition among individuals with hemorrhagic and ischemic stroke.

  1. Also, You talk of risk factors for stroke, not the factors associated with poorer or better stroke outcome- as You state in the title: "AFTER hemorrhagic and ischemic stroke" so term after has to be explained- when? 10d-? 30d-? 180d-?

 Response:  We have explained the term "after stroke". Qualification for examination was performed by a neurologist. He evaluated the patient's clinical condition and ability to assume a standing position. The inclusion criteria for the study were logical contact with the patient and the ability to maintain a standing position for at least 120 seconds. On average, the test usually took place 10 days after the stroke.

  1. You need to explain why the body composition is important for the stroke outcome more clearly.

Response:   It has been explained why body composition is of significance for risk of stroke.

  1. Line 65 -Thrombolysis was diagnosed in 11 patients (13%), while only one (4%) male was treated via mechanical thrombectomy.

Response: This sentenced was corrected as follows: Eleven (13%) patients were treated via thrombolysis, while only one (4%) were treated using mechanical trombectomy.

  1. In methods- as this study is done in rehabilitation center, then You should point out, what time after stroke or ICH was this performed.

The inclusion criteria were logical contact with the patient and the ability of each patient to maintain a standing position for at least 120 seconds. The study usually took place an average of 10 days following stroke onset. Patients were qualified for examination by a neurologist. He evaluated the patient's clinical condition and his ability to assume a standing position.

  1. In results the two separate tables are not necessary- just us the 3rd table, but ad the SD for each IS and ICH.

Response: Tables 1 and 2 have been deleted. In the remaining table, SD for each IS and ICH has been added.

  1. The title for 3rd table is incorrect.

Response: The title of the table has been changed to: Table 1. Differences in body composition between hemorrhagic and ischemic stroke.

  1. In methods as You are measuring certain things (all that are in your results)- You should explain what they are and what they mean, and why You consider them important.

Response: In the ‘Methods’ section, we have explained the significance of all the examined variables of body composition. Body mass (kg) (BM) means the total weight of the body in kilograms. Basal Metabolic Rate (kJ) (BMR) determines the minimal level of energy that the body needs to be able to function properly in resting state. Fat Percentage (%) (FP) is the percentage of body fat in total body mass. Fat Mass (kg) (FM) is the mass of fat tissue in kilograms. Fat-free mass (FFM) (kg) is the most important metabolic component of the body. It consists of muscle, internal organ and bone mass. Total Body Water (kg) (TBW) is the total fluid content of the body. Muscle Mass (kg) (MM) includes skeletal and smooth muscle mass. Impedance (Ω) (IMP) is a generalisation of electrical resistance. The electrical signal easily flows through tissues containing large amounts of water, e.g. muscle tissue. Passing through adipose tissue it encounters greater resistance because it contains small amounts of fluid. This resistance is called impedance. Impedance readings are then entered into medically established mathematical formulas to calculate body composition. Body Mass Index (BMI) (kg/m²) is the ratio of body mass to body height. This is the standard body mass index. Visceral Fat Level (VFL) means visceral fat located deep in the abdominal area. Bone mass (BM) (kg) is the mass of bone minerals in the body. Extracellular Water (kg) (ECW) denotes its content in the body. Intracellular Water (ICW) (kg) is its content in the body. Metabolic Age (MA) means metabolic rate. Genes, diet, physical activity, as well as hormonal balance affect metabolic age. In addition, this index also depends on BMR, body water content, body fat mass, muscle mass. Phase angle (PA) (°) determines overall health and functional the quality of tissues and cells. It is also an indicator of the body's nutrition. It is calculated from the resistance and reactance coefficient. Low phase angle may indicate physical exhaustion. It is also characteristic for malnourished and chronically ill individuals. The correct phase angle for women is >5°, while for men, it should be >6°.

  1. Also the results will differ if You will not use the left or right arm/leg instead i would suggest to use- paretic arms/legs nonparetic arms/legs, then You need to recalculate.

Response: We have introduced the following terms: paretic limb and non-paretic limb. Once more, we have calculated the means and standard deviation for these variables as well as the Anova – F.

  1. As You describe the results You should use the terms and abbreviations, and then just the abbreviations, no need to explain the kg etc. in discussion.

Response: This has been corrected, and in the ‘Discussion’ section, we have applied the same abbreviations.

  1. In discussion You should compare the results of Your study to other studies, not just tell what they have found, so You are not really discussing.

Response: The ‘Discussion’ section has been extensively edited and elaborated. We have compared the results of our research with those obtained in other studies.

  1. I don’t see an actual conclusion. Also here just use the abbreviations not the full text.

Response: We have corrected the ‘Conclusions’ section and have applied the abbreviations.

Significant differences in the body composition among patients after hemorrhagic and ischemic stroke have been indicated in this study. Individuals who had experienced ischemic stroke demonstrated significantly worse body composition parameters. Improper body composition is an important stroke risk factor – especially with regard to ischemic incidences.

Once more, we are exceptionally grateful for your in-depth review of our article. Your insight and comments will definitely allow for an increase in the substantive value of the manuscript. We hope that our detailed responses and the extensive changes to the text are sufficient for the publication of our text in your renowned journal. Thank you for your devoted time and effort.

Yours sincerely,

Assoc. Prof. UJK Jacek Wilczyński, Ph.D.

Round 2

Reviewer 3 Report

  1. I think this is incorrect or I would write it a bit differently:

In body composition, we may distinguish lean body mass (LBM), also called lean or adipose tissue

In the authors’ research- maybe use in the available research or in the published research, as You refer to some other studies

There were 11 (13%) patients treated with thrombolysis, while only 1 (4%) underwent mechanical thrombectomy. - I would write that % patients had received thrombolytic treatment with i/v tPA and mechanical thrombectomy.

Nonetheless, the results of this study provide great substantive value. Maybe use significant instead.

  1. In methods, You do not state that You compare the ischemic stroke patient and haemorrhagic stroke patients. 
  2. In the discussion, now You describe the other studies, without comparing them to Your results. 
  3. Also You need to state that, if You observed difference in some of the parameters, was it or was it not statistically significant?
  4. Significant differences in body composition of patients after haemorrhagic and ischemic stroke have been demonstrated. - I think You should state witch ones

Author Response

Response to comments by Reviewer

Thank you very much for your insightful comments and recommendations. We are very grateful for the time you have devoted to the review. We have implemented the required changes in the text. Below, please find a point-by-point response to the comments. Once more, thank you for the insight which, we are sure, has significantly improved the substantive value of our manuscript.

Comments and Suggestions for Authors I think this is incorrect or I would write it a bit differently:

 Response: This sentence was corrected according to the Reviewer’s suggestion and changed to: In body composition, we may distinguish lean body mass (LBM), also called lean or adipose tissue.

  • In the authors’ research- maybe use in the available research or in the published research, as You refer to some other studies.

Response: In all cases, this was corrected and changed to: available research or published research.

  • There were 11 (13%) patients treated with thrombolysis, while only 1 (4%) underwent mechanical thrombectomy.

Response: This sentence was corrected and changed to::  Thrombolytic treatment with i/v tPA and mechanical thrombectomy was received by 16% of patients.

  • There In theNonetheless, the results of this study provide great substantive value. Maybe use significant instead.

Response: This sentence was corrected and changed to: Nonetheless, the results of this study provide significant substantive value.

  • In methods, You do not state that You compare the ischemic stroke patient and haemorrhagic stroke patients.

Response: In the‘Materials and Methods’ section, it is stated that patients with hemorrhagic stroke are compared to those with ischemic stroke.We compared body composition of people after hemorrhagic and ischemic stroke.

  • In In the discussion, now You describe the other studies, without comparing them to Your results.

Response: The ‘Discussion’ section has been corrected. Other studies have also been described and commented on, comparing them to the results fo our reserach.

  • In Also You need to state that, if You observed difference in some of the parameters, was it or was it not statistically significant?

Response: If differences were noted in any of the parameters, information is always provided as to whether they were of statistical significance or not.

  • Significant differences in body composition of patients after haemorrhagic and ischemic stroke have been demonstrated. - I think You should state witch ones.

Response:

Significant differences in body composition of patients after hemorrhagic and ischemic stroke have been demonstrated. Individuals after ischemic stroke had significantly worse body composition. In ischemic strokes, significantly higher values of body composition variables compared to hemorrhagic strokes concerned BM, BMR (kJ), FFM (kg), TBW (kg), MM (kg), VFL, BNM (kg), ECW (kg), ICW (kg), TFFM (kg), TMM (kg), FFM (kg) and MM (kg) in the upper limb with paresis, FFM (kg) and MM (kg) in the non-paretic upper limb, FFM (kg) and MM (kg) in the lower limb with paresis, FFM (kg) and MM (kg) in the lower limb without paresis. Incorrect body composition is a significant risk factor, especially of ischemic stroke.

Once more, we are exceptionally grateful for your in-depth review of our article. Your insight and comments will definitely allow for an increase in the substantive value of the manuscript. We hope that our detailed responses and the extensive changes to the text are sufficient for the publication of our text in your renowned journal. Thank you for your devoted time and effort.

Yours sincerely,

Assoc. Prof. UJK Jacek Wilczyński, Ph.D.
